# Metabolic Characteristics of Gut Microbiota and Insomnia: Evidence from a Mendelian Randomization Analysis

**DOI:** 10.3390/nu16172943

**Published:** 2024-09-02

**Authors:** Fuquan Xie, Zhijun Feng, Beibei Xu

**Affiliations:** 1Institute of Biomedical & Health Engineering, Shenzhen Institute of Advanced Technology, Chinese Academy of Sciences, Shenzhen 518055, China; fuquanxie@163.com; 2Department of Radiation Medicine, Guangdong Provincial Key Laboratory of Tropical Disease Research, School of Public Health, Southern Medical University, Guangzhou 510515, China; fengzhj18@lzu.edu.cn; 3Institute of Synthetic Biology, Shenzhen Institute of Advanced Technology, Chinese Academy of Sciences, Shenzhen 518055, China

**Keywords:** insomnia, gut microbiota pathway, Mendelian randomization, metabolic pathways, causal relationship

## Abstract

Insomnia is a common sleep disorder that significantly impacts individuals’ sleep quality and daily life. Recent studies have suggested that gut microbiota may influence sleep through various metabolic pathways. This study aims to explore the causal relationships between the abundance of gut microbiota metabolic pathways and insomnia using Mendelian randomization (MR) analysis. This two-sample MR study used genetic data from the OpenGWAS database (205 gut bacterial pathway abundance) and the FinnGen database (insomnia-related data). We identified single nucleotide polymorphisms (SNPs) associated with gut bacterial pathway abundance as instrumental variables (IVs) and ensured their validity through stringent selection criteria and quality control measures. The primary analysis employed the inverse variance-weighted (IVW) method, supplemented by other MR methods, to estimate causal effects. The MR analysis revealed significant positive causal effects of specific carbohydrate, amino acid, and nucleotide metabolism pathways on insomnia. Key pathways, such as gluconeogenesis pathway (GLUCONEO.PWY) and TCA cycle VII acetate producers (PWY.7254), showed positive associations with insomnia (*B* > 0, *p* < 0.05). Conversely, pathways like hexitol fermentation to lactate, formate, ethanol and acetate pathway (P461.PWY) exhibited negative causal effects (*B* < 0, *p* < 0.05). Multivariable MR analysis confirmed the independent causal effects of these pathways (*p* < 0.05). Sensitivity analyses indicated no significant pleiotropy or heterogeneity, ensuring the robustness of the results. This study identifies specific gut microbiota metabolic pathways that play critical roles in the development of insomnia. These findings provide new insights into the biological mechanisms underlying insomnia and suggest potential targets for therapeutic interventions. Future research should further validate these causal relationships and explore how modulating gut microbiota or its metabolic products can effectively improve insomnia symptoms, leading to more personalized and precise treatment strategies.

## 1. Introduction

Insomnia, a prevalent sleep disorder, affects the sleep quality and daily life of millions worldwide [1]. According to the World Health Organization, approximately 10% to 30% of adults suffer from insomnia, with a higher prevalence among the elderly and those with health conditions [2]. Insomnia is not just a nighttime issue; it is associated with various physical and mental health problems, including cardiovascular diseases [3,4,5], diabetes [2,6,7], depression [8,9,10,11], and anxiety [12,13]. Additionally, insomnia has significant socioeconomic impacts, such as reduced work efficiency [14], increased accident risk, and higher healthcare costs [15].

The gut microbiota, commonly known as the gut microbiome, comprises thousands of microbial species residing in the human gastrointestinal tract [16,17]. These microbes not only aid in digestion but also influence the host’s immune system, metabolism, and neural functions [18,19]. Recent research [20,21,22,23,24,25,26] indicates that the gut microbiota affects human behavior and emotions through the “gut–brain axis”, particularly in regulating stress responses, mood, and sleep cycles. For example, certain gut bacteria produce neurotransmitters related to brain function, such as serotonin [27,28] and gamma-aminobutyric acid (GABA) [29,30], which can directly or indirectly influence sleep and mood states. While existing studies have explored the association between the diversity of gut microbiota and various health conditions, they primarily focus on the types and quantities of microbial species. For instance, research has shown that individuals with insomnia have lower gut microbiota diversity, potentially linked to mechanisms of emotional regulation and stress response [31,32]. Additionally, specific gut bacteria, such as Faecalibacterium [32,33,34] and Bifidobacterium [35,36,37], have been associated with better sleep quality. However, studies on gut bacterial pathways are relatively scarce. The study of the abundance of gut bacterial pathways primarily focuses on understanding the metabolic characteristics of the microbiota, including the synthesis of various bioactive molecules such as lipids, carbohydrates, and amino acids [38,39,40]. These metabolites are crucial in regulating a host of physiological functions within the body, including immune responses, inflammation control, and energy metabolism. By elucidating these interactions, researchers can explore how alterations in the gut microbiome may influence overall health and disease states, providing insights into potential therapeutic strategies that target microbiota to improve health outcomes. For example, SCFAs, produced by gut bacteria fermenting food residues, possess anti-inflammatory and protective properties and have been shown to directly influence brain function and behavior through the blood–brain barrier [41,42]. By studying the abundance of gut bacterial pathways, we can more accurately grasp the common physiological roles of one or more gut microbiota, thereby improving our understanding of these functional microbial communities. Additionally, this research helps us gain a deeper understanding of how microorganisms affect hosts through specific chemical signals. This approach provides potential theoretical support for future target identification based on gut microbiota-related metabolic characteristics. In other words, the study of gut microbiota pathways is not limited to analyzing microbial species themselves but primarily focuses on how microorganisms influence host life activities through their metabolites. This method offers a new perspective on how microorganisms regulate the host’s physiological and pathological processes, helping to uncover mechanisms for improving health by modulating gut microbiota and its related metabolic pathways.

Mendelian randomization (MR) utilizes genetic variation as an instrumental variable, offering a robust strategy to assess causal relationships and effectively overcoming the limitations of traditional observational studies [43]. This study hypothesizes that specific gut microbiome pathways may have a potential causal relationship with insomnia. Specifically, we expect that, first, single nucleotide polymorphisms (SNPs) closely associated with the abundance of gut microbiome pathways may influence the risk of insomnia. Second, certain microbial pathways (such as those related to neurotransmitter production or metabolism) may have stronger associations with insomnia risk than others. Third, these associations may be realized by altering the function of the gut–brain axis or influencing metabolic processes related to sleep regulation (such as gluconeogenesis). To test these hypotheses, this study utilized genome-wide association study (GWAS) data closely related to the abundance of gut microbiota pathways, employing MR methods to explore the potential causal relationships between genetic variations associated with microbial pathways and insomnia. The innovation of this approach lies in its ability to directly reveal the causal links between the metabolic characteristics of gut microbiota and human sleep behavior. By analyzing the causal effects of specific gut bacteria-related biological pathways on sleep regulation, this study not only provides biomarkers and potential targets for developing new treatment strategies for insomnia but also deepens our understanding of the gut–brain axis in maintaining health and disease states. We hope this research will offer new perspectives for clinical treatments and guide future health interventions, ultimately enhancing people’s quality of life.

## 2. Materials and Methods

### 2.1. Data Sources

This study is a bidirectional two-sample MR analysis conducted according to the STROBE-MR guidelines (Strengthening the Reporting of Mendelian Randomization Studies) [44]. We used data from the OpenGWAS database (https://gwas.mrcieu.ac.uk/, accessed on 10 July 2024) on the abundance of 205 gut bacterial pathways (ebi-a-GCST90027446 to ebi-a-GCST90027650, sample size: 7738) as exposure variables [45]. Insomnia data from the R10 version of the FinnGen database (finngen_R10_F5_INSOMNIA, case: 4801, control: 405,229) served as outcome variables [46]. According to MR guidelines [47], we selected exposure and outcome data from European populations but from different regions, choosing datasets with the largest sample sizes or the highest number of cases for this analysis. Through this approach, we aim to reveal the causal relationship between specific gut microbiota-related pathways and insomnia, providing potential biomarkers or targets for the prevention and treatment of insomnia based on gut microbiome levels. Figure 1 illustrates the overall design and framework of this study.

### 2.2. Instrumental Variable (IV) Selection

#### 2.2.1. Selection of Exposure-Related IVs

We employed the “TwoSampleMR” R package to perform a two-sample MR analysis [48], aiming to identify SNPs closely related to the exposure factors, using these SNPs as instrumental variables (IVs). In obtaining IVs, we set parameters including a *p*-value (statistical significance threshold) of 2 × 10^−5^, an LD threshold (linkage disequilibrium level) of 0.001, and a window size of 10,000 kilobase pairs [49,50]. In this study, we use the term “causal effect” to describe the estimated impact of genetic variants associated with gut microbiome pathways on insomnia risk, as inferred through MR analysis. Specifically, a “causal effect” in this context suggests that variation in these genetic markers, which are proxies for gut microbiome pathway abundance, is associated with changes in insomnia risk. However, it is important to note that while MR provides evidence for potential causal relationships, it is subject to certain assumptions and limitations. In other words, the selected IVs must satisfy three core MR assumptions [43]: relevance assumption—SNPs are strongly associated with the exposure factor; independence assumption—SNPs are not associated with potential confounders; exclusion restriction assumption—SNPs influence the outcome solely through the exposure factor. We validated the IVs’ effectiveness through quality control, ensuring completeness in sample size and effect allele frequency (EAF). For missing sample size data, we supplemented with databases like “openGWAS”; for incomplete EAF information, we used data from the “1000 Genomes Project” [51].

#### 2.2.2. Removing Confounding IVs

To eliminate potential confounding IVs, we analyzed the LDtrait tool (version: LDlink 5.6.8) from the “LDlink” database (https://ldlink.nih.gov/?tab=ldtrait, accessed on 12 July 2024) [52,53]. This tool assesses the associations of specific SNPs with various phenotypes. If an IV is significantly associated with sleep-related phenotypes (e.g., sleep disorders, sleep disturbances, sleep apnea), it is identified as a potential confounder and excluded. Subsequently, we calculated the F-statistics (F = beta2/se2) for the remaining IVs [54,55], ensuring all F-values exceed 10, which helps maintain the statistical power and strength of the IVs in the MR analysis [56]. We then integrated IVs related to both exposure and outcome, ensuring their effect alleles matched the reference alleles and used the “RadialMR” R package in conjunction with the “MR-PRESSO” R package to identify and remove outlier IVs [57,58]. This data-cleaning step yielded the final IVs for the MR analysis.

### 2.3. MR Analysis

The MR analysis was primarily conducted using the “TwoSampleMR” R package [48], designed specifically for performing two-sample MR analyses. We used the inverse variance-weighted (IVW) method as the main approach [59], supplemented by MR Egger [60], weighted median [61], simple mode [62], and weighted mode methods for further validation [63]. Significant results were based on an IVW method *p*-value of less than 0.05, with consistent beta values across other supplementary methods [59,64]. Using the “TwoSampleMR” R package, we calculated the odds ratios (OR) and their 95% confidence intervals (CI) for each direction of analysis, thus assessing the causal effects revealed by the MR analysis.

### 2.4. Multivariable MR (MVMR) Analysis

In analyzing the causal relationship between different types of gut bacterial pathway abundances and insomnia, we applied the “mvmr” R package for MVMR analysis to the data showing significant causal relationships [65]. This aimed to evaluate the independent causal effects of different types of gut bacterial pathways on insomnia. These steps enhanced the rigor of the study methods and the reliability of the results, furthering our understanding of the causal relationship between gut bacterial pathway abundance and insomnia and providing a theoretical basis for exploring potential pathological mechanisms.

### 2.5. Sensitivity Analysis

The primary goal of the sensitivity analysis was to explore whether IVs were affected by horizontal pleiotropy and heterogeneity [66]. For instances involving more than two SNPs, we used Cochran’s Q test to assess heterogeneity [67]. A *p*-value for a Q test less than 0.05 indicated significant heterogeneity, necessitating the use of the “random-effects model-IVW” method for MR analysis to accommodate IV inconsistency [68,69]. To detect horizontal pleiotropy, we used the MR Egger intercept method if the number of SNPs exceeded two and the MR-PRESSO global test if it exceeded four [57,70]. We also visualized sensitivity analysis results using scatter plots and funnel plots, which helped reveal data distribution and potential biases. Additionally, we conducted a leave-one-out analysis, removing each IV one at a time and recalculating MR results to assess each IV’s contribution to the overall outcome. These methods ensured the study’s rigor and the reliability of the results, deepening our understanding of the relationship between gut bacterial pathway abundance and insomnia and providing significant scientific support for precision medicine.

## 3. Results

### 3.1. Causal Effects of Different Types of Gut Bacterial Pathway Abundance on Insomnia

According to MR guidelines, the GWAS datasets included in this analysis (205 types of gut bacterial pathway abundance from the OpenGWAS database and one GWAS dataset related to insomnia patients from the FinnGen database) are provided in Appendix A. We first screened SNPs closely related to the exposure variables (205 types of gut bacterial pathway abundance). After excluding potential confounders, we selected SNPs with F-values greater than 10 as IVs for this analysis to ensure their rationality and effectiveness in MR analysis. 

The MR results, using the IVW as the primary analysis method, showed that nine types of gut bacterial pathway abundance had significant causal relationships with insomnia (Figure 2, *p* < 0.05). Among these, pathways related to carbohydrate metabolism, such as gluconeogenesis pathway (GLUCONEO.PWY, ebi-a-GCST90027473) and starch degradation pathway (PWY.6731, ebi-a-GCST90027597), amino acid metabolism pathways, such as super pathway of L-isoleucine biosynthesis pathway (PWY.3001, ebi-a-GCST90027531) and 8-amino-7-oxononanoate biosynthesis pathway (PWY.6519, ebi-a-GCST90027582), nucleotide metabolism pathways, such as adenine and adenosine salvage pathway (PWY.6609, ebi-a-GCST90027587), and other metabolism-related pathways, such as tricarboxylic acid (TCA) cycle VII acetate producers pathway (PWY.7254, ebi-a-GCST90027619), showed positive causal effects on insomnia (*B* > 0, *P*_ivw_ < 0.05, Figure 2). Conversely, pathways related to carbohydrate metabolism, such as hexitol fermentation to lactate, formate, ethanol and acetate pathway (P461.PWY, ebi-a-GCST90027503), other metabolism-related pathways, such as tRNA processing pathway (PWY0.1479, ebi-a-GCST90027522), and cell wall biosynthesis pathways, such as peptidoglycan maturation meso diaminopimelate containing pathway (PWY0.1586, ebi-a-GCST90027524), showed negative causal effects on insomnia (*B <* 0, *P*_ivw_ < 0.05, Figure 2). Detailed information on the MR analysis results in each direction is provided in Appendix A. 

### 3.2. MVMR Analysis

We used MVMR analysis to evaluate the independent causal effects of gut bacterial pathway abundances that showed significant associations in the univariate MR analysis on insomnia. The results (Figure 3) revealed that the gluconeogenesis pathway (GLUCONEO.PWY) and the TCA cycle (PWY.7254) have positive causal relationships with insomnia (*B* > 0, *P*_ivw_ < 0.05, Figure 3), suggesting that the activity of these pathways may reflect certain abnormalities in energy metabolism, potentially contributing to or exacerbating insomnia symptoms. Additionally, the polyol fermentation pathway (P461.PWY) exhibited a negative causal relationship with insomnia (*B* < 0, *P*_ivw_ < 0.05, Figure 3), indicating that this pathway might play a protective role in maintaining normal sleep patterns by reducing metabolic burden or regulating energy production. Detailed results of the MVMR analysis are provided in Appendix A. These findings confirm the independent roles of the three key metabolic pathways in the development of insomnia and provide new research directions for improving or preventing insomnia by modulating specific gut bacterial metabolic pathways.

### 3.3. Sensitivity Analysis

In the forward analysis, no significant pleiotropy or heterogeneity was observed in those directions with significant MR analysis (Appendix A). For the nine gut bacterial pathway types that showed significant associations in the univariate MR analysis, individual SNPs displayed notable differences in their causal effect estimates on insomnia (Appendix A). Subsequently, we conducted a leave-one-out analysis, sequentially removing each SNP in each direction and recalculating the MR results. The results (Figure 4A–I) indicated that the removal of individual SNPs only affected the range of maximum and minimum causal effect estimates but had minimal impact on the overall direction of the causal effect estimates. This suggests that despite the significant differences in individual SNPs in various directions, these differences may contribute to the complexity of gut microbiota-related metabolic pathways involved in the progression of insomnia. However, from an overall perspective, the causal effect estimates of these SNPs on insomnia are consistent, reinforcing the significant role of gut microbiota-related metabolic pathways in the progression of insomnia.

Scatter plots for the univariate MR analysis (Figure 5A–I) display the effect sizes of the SNPs on the exposure and outcome variables, fitting a potential linear trend. The presence of this linear trend with a slope is consistent with our MR analysis results, indicating a significant causal relationship between exposure and outcome in each direction. Funnel plots (Appendix A) also demonstrate variability in the distribution of SNPs across different MR analysis methods in each direction, but no significant distribution anomalies were observed when using the IVW method as the primary approach. This evidence further confirmed the robustness of the analysis. Overall, the sensitivity analysis results indicate that our study results are highly reliable and robust, supporting the important causal role of gut bacterial pathway abundance in the development of insomnia.

## 4. Discussion

This study utilized MR to uncover the causal relationships between the abundance of gut microbiota pathways and insomnia. The results of the univariable two-sample MR analysis indicated that pathways related to carbohydrate metabolism, such as gluconeogenesis pathway (GLUCONEO.PWY) and starch degradation pathway (PWY.6731), amino acid metabolism pathways such as super pathway of L-isoleucine biosynthesis pathway (PWY.3001) and 8-amino-7-oxononanoate biosynthesis pathway (PWY.6519), nucleotide metabolism pathways such as adenine and adenosine salvage pathway (PWY.6609), and other metabolism-related pathways such as TCA cycle VII acetate producers pathway (PWY.7254) had positive causal effects on insomnia. Conversely, pathways related to carbohydrate metabolism, such as hexitol fermentation to lactate, formate, ethanol and acetate pathway (P461.PWY), other metabolism-related pathways, such as tRNA processing pathway (PWY0.1479), and cell wall biosynthesis pathways, such as peptidoglycan maturation meso diaminopimelate containing pathway (PWY0.1586) had negative causal effects on insomnia. The results of the MVMR analysis confirmed the independent positive causal effects of pathways related to carbohydrate metabolism (GLUCONEO.PWY, gluconeogenesis I) and other metabolism-related pathways (PWY.7254, TCA cycle VII acetate producers) on insomnia, as well as the independent negative causal effect of the carbohydrate metabolism pathway (P461.PWY, hexitol fermentation to lactate, formate, ethanol and acetate) on insomnia. The evidence found in the study supports the hypothesis that specific metabolic pathways, such as gluconeogenesis (GLUCONEO.PWY) and the TCA cycle pathways (PWY.7254), may play active roles in the development of insomnia. These metabolic pathways may affect sleep quality by influencing energy balance and the accumulation of metabolic products, providing new insights into the role of energy metabolism in sleep regulation. The sensitivity analysis results showed no significant pleiotropy or heterogeneity in any direction of the MR analysis. The leave-one-out analysis indicated that removing individual SNPs had minimal impact on the overall causal effect estimates, further ensuring the robustness of the analysis results. In summary, this study revealed the complex relationship between specific metabolic pathways in the gut microbiota and insomnia, opening new avenues for exploring novel therapeutic strategies for insomnia. These findings provide new insights into the role of the gut microbiota in regulating physiological functions and offer a theoretical basis for developing interventions targeting specific metabolic pathways. By modulating these key metabolic pathways, we hope to improve the sleep quality of insomnia patients, thereby enhancing their quality of life and bringing hope to a broad population of insomnia sufferers.

From a clinical perspective, this study has practical implications. Firstly, it reveals the causal relationships between specific metabolic pathways and insomnia, providing targeted prevention strategies for diagnosing and treating insomnia. The positive causal effect of pathways such as gluconeogenesis and the tricarboxylic acid cycle, as well as the inverse causal role of the polyol fermentation pathway on insomnia, provide scientific evidence for developing new therapeutic approaches. For example, regulating the activity of the gluconeogenesis and tricarboxylic acid cycle pathways can reduce the risk of insomnia by modulating energy metabolism [71,72]. Meanwhile, enhancing the activity of the polyol fermentation pathway may help maintain energy balance and protect normal sleep patterns [73]. This evidence also suggests that to reduce and prevent the occurrence of insomnia, it is advisable to decrease the intake of high-sugar and high-fat foods before sleep or during the bedtime period and appropriately increase the intake of foods rich in protein and fiber to ensure that energy metabolism remains relatively stable, thereby safeguarding sleep quality and reducing the risk of insomnia. Secondly, this study offers new targets for drug development for insomnia. Specific metabolic pathways identified in the study, such as gluconeogenesis and TCA cycle VII acetate producers, can serve as potential targets for developing novel insomnia treatments. Drugs targeting these pathways may regulate energy balance and metabolic processes, providing a new method for treating insomnia. Finally, this study provides a practical foundation for implementing precision medicine. Supporting the theory that metabolic pathways mediated by the gut microbiota influence insomnia allows clinicians to analyze patients’ gut microbiota to identify metabolic pathway abnormalities and formulate personalized treatment plans accordingly. For example, adjusting the diet or using specific probiotic supplements to influence certain metabolic pathways could regulate patients’ sleep quality [35,74,75]. These clinical implications not only deepen our understanding of the mechanisms underlying insomnia but also offer more targeted and predictive treatment methods that could significantly improve patients’ quality of life and therapeutic outcomes.

From a metabolic perspective, insomnia is also considered a disease closely related to energy metabolism, emphasizing metabolic imbalance as a potential factor causing insomnia [76]. Specifically, abnormalities in energy metabolism can affect sleep regulation mechanisms through various pathways, leading to insomnia. Firstly, fluctuations in energy metabolism can directly impact brain function, particularly brain regions closely related to sleep regulation. For example, glucose metabolism abnormalities can affect the energy supply to the cerebral cortex and hypothalamus, both crucial in maintaining the sleep–wake cycle [77,78]. Research has shown that reduced glucose utilization can impact neural activity in these brain regions, potentially leading to changes in sleep structure, such as delayed sleep or insufficient sleep [79,80]. Secondly, disorders in energy metabolism can further interfere with sleep by affecting hormone balance [81,82,83]. Insulin and cortisol are two crucial hormones regulating energy metabolism, and their abnormal secretion is closely related to sleep quality [84,85,86]. Insulin resistance is usually accompanied by a decline in sleep quality [87,88,89,90], while high levels of nighttime cortisol are associated with sleep disruptions and early awakening [91,92,93]. Therefore, dysregulation of metabolism-related hormones affects the body’s energy state and may also impact the brain’s sleep regulation system. Additionally, metabolic disorders are often accompanied by chronic low-grade inflammation [94,95], and inflammatory factors such as interleukins and tumor necrosis factors have been shown to penetrate the blood–brain barrier [96], affecting brain neural activity, including sleep regulation. Elevated levels of inflammatory factors can activate the cerebral cortex and limbic system, increasing the frequency of awakenings and reducing the proportion of deep sleep [97,98,99]. In conclusion, understanding insomnia from a metabolic perspective helps reveal the disease’s complex mechanisms and provides new insights for treatment. Adjusting metabolic status may positively impact improving sleep quality, including regulating energy metabolism through diet, exercise, and medication to optimize sleep state.

In this analysis, we identified nine metabolic pathways closely related to gut microbiota, mainly involving carbohydrate metabolism, amino acid metabolism, and nucleotide metabolism. Notably, carbohydrate metabolism pathways play a complex dual role in the development of insomnia. For example, the gluconeogenesis pathway (GLUCONEO.PWY, gluconeogenesis I) and the starch degradation pathway (PWY.6731, starch degradation III) are generally associated with enhanced energy metabolism. These pathways are active in gut microbiota such as Escherichia coli [100,101] and Clostridium [102,103], and by increasing nighttime blood glucose levels or influencing blood glucose fluctuations, they may negatively affect sleep. On the other hand, the pathway of hexitol fermentation to lactate, formate, ethanol, and acetate (P461.PWY) mainly involves the metabolic activities of acid-producing bacteria such as Bifidobacterium and Lactobacillus. These bacteria produce lactic acid and other beneficial metabolic products through fermentation, potentially regulating nervous system function and promoting sleep [104,105,106,107,108]. Therefore, reducing the intake of carbohydrates and starches before bedtime and supplementing with fermented foods containing acid-producing probiotics such as yogurt may help regulate the balance of gut microbiota, optimize the activity of related metabolic pathways, and reduce insomnia. 

Adenosine, an important energy molecule and neuromodulator [109,110,111], is synthesized and recycled mainly through the adenine and adenosine salvage III pathway (PWY.6609). Adenosine plays a central role in regulating the sleep–wake cycle, especially in promoting sleep [112,113,114]. Typically, we expect enhanced adenosine synthesis to promote sleep, as adenosine inhibits the central nervous system in sleep regulation [115,116]. However, our analysis observed a positive causal effect between the nucleotide metabolism pathway (PWY.6609, adenine and adenosine salvage III) and insomnia. This may indicate several scenarios. First, the adaptive response of adenosine accumulation [117,118]. In insomnia patients, there may be adaptive enhancement of adenosine metabolism pathways as a compensatory mechanism. That is, due to poor sleep quality, the body may attempt to promote sleep by increasing adenosine production, but this increase may still be insufficient to overcome other factors causing insomnia. Second, decreased adenosine sensitivity [119]. Insomnia patients may have reduced sensitivity to adenosine, meaning that even with increased adenosine levels, its sleep-promoting effects may not be significant. This decreased sensitivity may be due to changes in receptors or other signaling pathways regulating adenosine’s action. Third, abnormal activation of metabolic pathways. In some cases, increased adenosine synthesis may reflect broader abnormalities in energy metabolism, which may be related to the pathological state of insomnia [120,121]. Therefore, while intuitively, increased adenosine should be associated with sleep promotion, this relationship may be influenced by various physiological and pathological factors in the complex context of insomnia, leading to the observed positive causal relationship. This suggests that future research needs to explore the specific roles and mechanisms of adenosine metabolism in insomnia more deeply.

In this analysis, we also identified the positive causal effects of L-isoleucine biosynthesis super pathway I (PWY.3001) and 8-amino-7-oxononanoate biosynthesis I pathway (PWY.6519) on insomnia. The connection between the activation of these metabolic pathways and the development of insomnia can be explained through several biochemical mechanisms. First, neurotransmitter synthesis. L-isoleucine is a precursor for the synthesis of various neurotransmitters, particularly in producing key excitatory neurotransmitters such as dopamine and norepinephrine [122,123]. These neurotransmitters play a central role in regulating mood, stress response, and the sleep–wake cycle. Enhanced biosynthesis pathways may lead to the overproduction of these neurotransmitters, disrupting normal sleep patterns. Second, energy metabolism and cellular stress response. 8-amino-7-oxononanoate is a crucial intermediate metabolite involved in a series of complex biochemical reactions, including those related to energy production [124,125]. The activation of this pathway may reflect the metabolic state of cells responding to oxidative stress or increased energy demand. Insomnia may be related to changes in energy metabolism, which may further affect sleep by impacting the energy balance of cells. Overall, the activation of these metabolic pathways may be the body’s attempt to adapt to internal or external environmental stress. However, excessive activation of these pathways may lead to biological rhythm disorders, ultimately affecting sleep quality. These findings provide new clues for further exploring the relationship between metabolic regulation and insomnia and may help develop treatments targeting these pathways to improve sleep conditions in insomnia patients.

The TCA cycle, also known as the citric acid cycle or Krebs cycle, is a crucial energy-producing process within cells, occurring in the mitochondria [126,127]. This cycle is one of the main pathways for aerobic cells to produce energy, especially when burning carbohydrates, fats, and proteins [128,129]. The significance of the TCA cycle lies not only in its role in energy production but also in providing precursor substances for many biosynthetic processes, such as the biosynthesis of certain amino acids [130,131]. Therefore, the TCA cycle is a core part of cellular metabolism, affecting nearly all biological energy and material metabolic processes. In the current analysis, the confirmation of the positive causal influence of TCA cycle acetate producers (PWY.7254) indicates a complex role of the TCA cycle in insomnia. The impact of this metabolic pathway on insomnia can be understood through several biochemical and physiological mechanisms. First, the effect of the TCA cycle on energy metabolism and brain function [132,133,134]. In the brain, the efficiency of the TCA cycle directly affects the energy supply and function of nerve cells, including the synthesis and release of neurotransmitters. If the TCA cycle is enhanced, it may lead to excessive brain energy metabolism, which in some cases may be associated with insomnia symptoms. Second, the impact of metabolic products. Metabolic products of the TCA cycle, such as acetate, may have the potential to regulate brain neural activity. Acetate, as a tissue hormone, can influence behavior and physiological states by affecting neural cells’ activity in certain situations [135,136,137]. Third, oxidative stress and sleep regulation. Enhanced TCA cycle activity may lead to changes in cellular oxidative stress levels, which have complex interactions with sleep regulation [138,139,140]. Excessive free radicals and oxidants may negatively affect the nervous system, including the neurotransmitter system and brain regions involved in sleep regulation. 

The current analysis employed the MR method, which attempts to avoid confounders and environmental interference by utilizing genetic variation as instrumental variables, enhancing the accuracy of causal inference. During implementation, we strictly adhered to the core principles of MR analysis, carefully selecting and quality-controlling IVs, and combined various statistical methods to analyze the data to ensure the reliability and robustness of the study results. These measures provide a solid scientific basis for the practical value of this study. However, there are also limitations to this study. First, the current analysis only included a sample size of over a thousand insomnia patients, and the sample size limitation may affect the broad applicability and reproducibility of the study results. Second, despite the stringent selection criteria and quality control steps, any research method or tool may have inherent limitations and unidentified potential confounders and biases, which may bring unpredictable impacts to the current analysis results and ultimately affect the accuracy and interpretability of the results. Third, the etiology of insomnia is complex, involving the interaction of genetic, environmental, physiological, and psychological factors. Although this study provides new insights from the perspective of gut microbiota metabolic pathways, this is only a part of the extensive factors. Therefore, these findings may not fully explain all biological mechanisms or pathological changes of insomnia and may limit our understanding of the broader causal relationships of the disease. Future research needs to adopt more comprehensive approaches, considering multiple biomarkers and environmental factors, to deeply explore and explain the complex etiological network of insomnia and support the development of more effective prevention and treatment strategies.

## 5. Conclusions

Although removing individual SNPs in the current MR analysis may affect the range of effect estimates, the overall trend of the MR analysis remains stable, providing robust evidence for our findings. The results indicate that key pathways such as carbohydrate metabolism, amino acid metabolism, and nucleotide metabolism demonstrated significant positive associations with the development of insomnia. These findings provide new perspectives for understanding the complex biological mechanisms of insomnia and offer theoretical bases for developing interventions targeting specific metabolic pathways. By uncovering the roles of these key metabolic pathways, this study not only enhances our understanding of the biological foundations of insomnia but also suggests potential targets for future therapeutic strategies, bringing new hope to insomnia patients. Future research needs to further validate these causal relationships and explore how adjusting the gut microbiota or its metabolic products can effectively improve insomnia symptoms, achieving more personalized and precise treatment methods.

## Figures and Tables

**Figure 1 nutrients-16-02943-f001:**
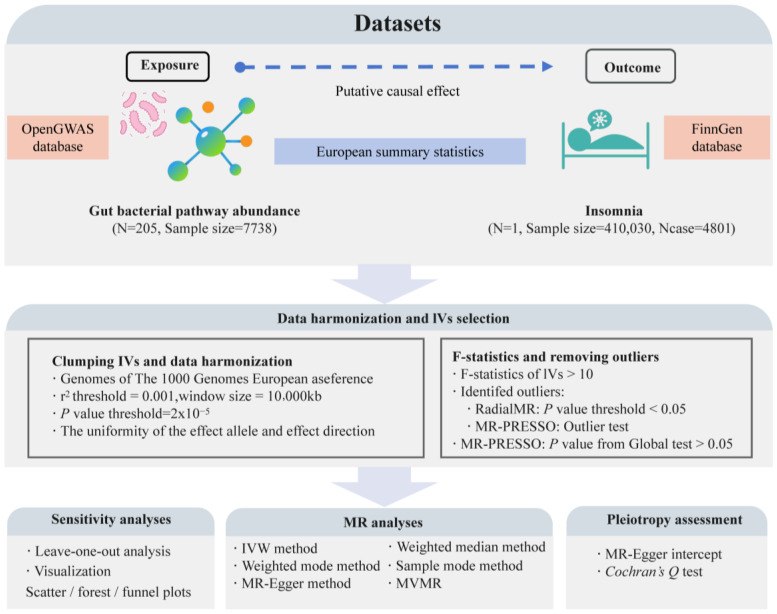
The flowchart of the study design and the data analysis steps. N represents the number of genome-wide association study (GWAS) datasets; Ncase represents the sample size with insomnia; MR, Mendelian randomization; IVW, inverse-variance weighted; MVMR, multivariable MR.

**Figure 2 nutrients-16-02943-f002:**
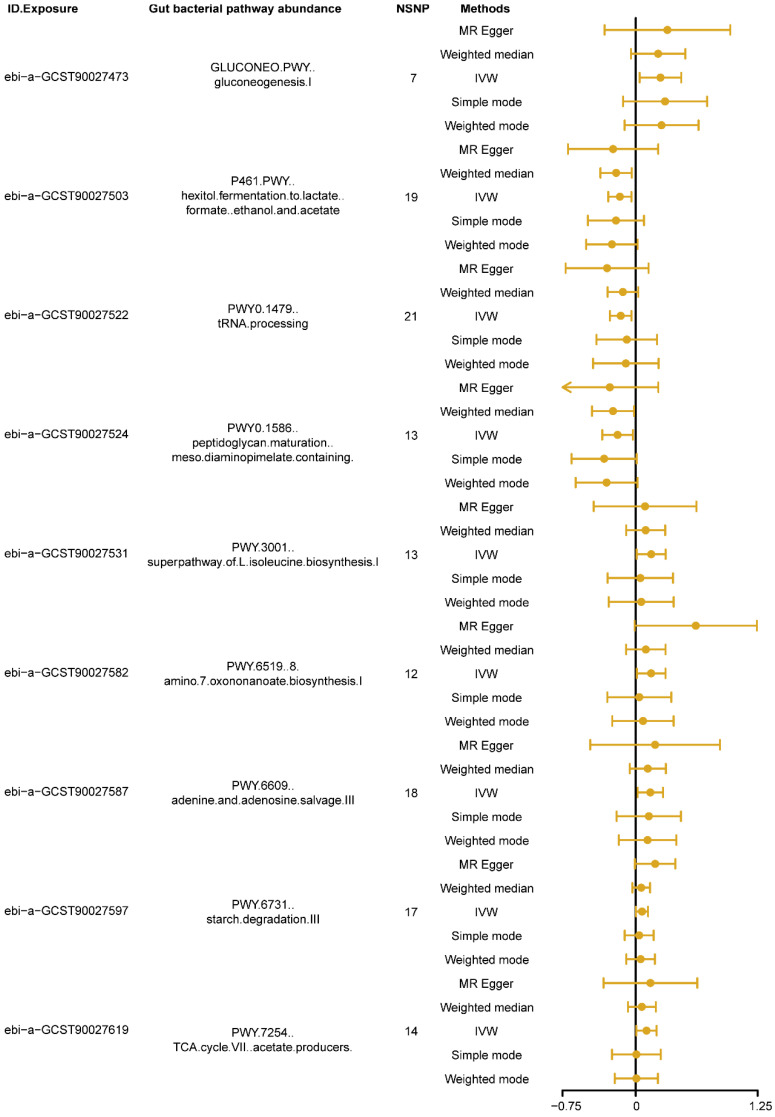
The forest plot shows the significant results of Mendelian randomization (MR) analysis between gut bacterial pathway abundance and insomnia. The dots represent the overall causal estimation in the analysis direction with different MR methods; the horizontal lines represent the upper (right) and lower (left) limits of effect estimation. ID.Exposure, the identification number of data set in OpenGWAS database; NSNP, number of single nucleotide polymorphism; IVW, inverse-variance weighted.

**Figure 3 nutrients-16-02943-f003:**
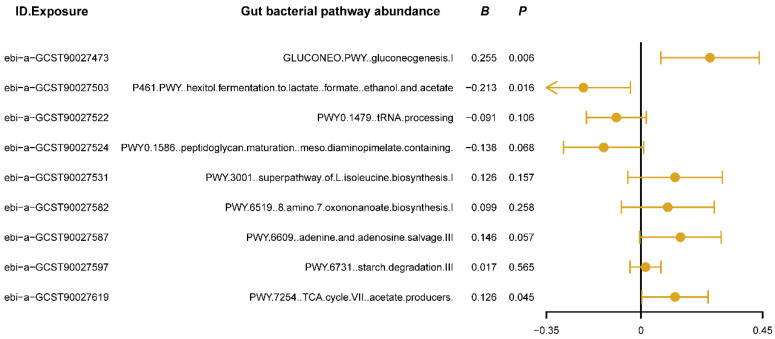
The forest plot shows the results of multivariable Mendelian randomization (MVMR) analysis between gut bacterial pathway abundance and insomnia. The dots represent the overall causal estimation in the analysis direction with “mvmr” R package, the horizontal lines represent the upper (right) and lower (left) limits of effect estimation. ID.Exposure, the identification number of data set in OpenGWAS database; *B*, the value of causal estimation; *p*, *p* value.

**Figure 4 nutrients-16-02943-f004:**
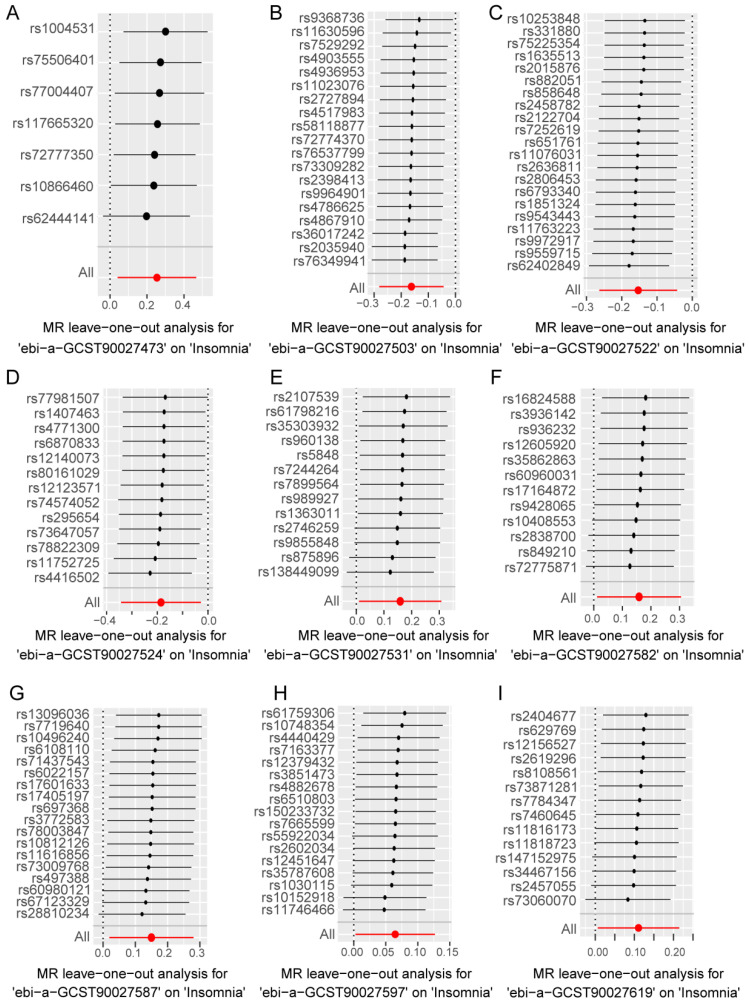
Leave-one-out analysis to assess the effect of each SNP in driving causality. (**A**–**I**) represents different analysis directions, and the specific directions are explained in the titles directly below each figure. The black dots represent the causal estimation effect size of the remaining SNPs after removing the current SNP in the current analysis direction, and the black lines with black dots represent the range of variation between the maximum (right end) and minimum (left end) values of this effect. The red dot represents the causal estimation effect of all SNPs, and the red line passing through the dot represents the range of variation of the maximum (right end) and minimum (left end) values of the overall causal effect.

**Figure 5 nutrients-16-02943-f005:**
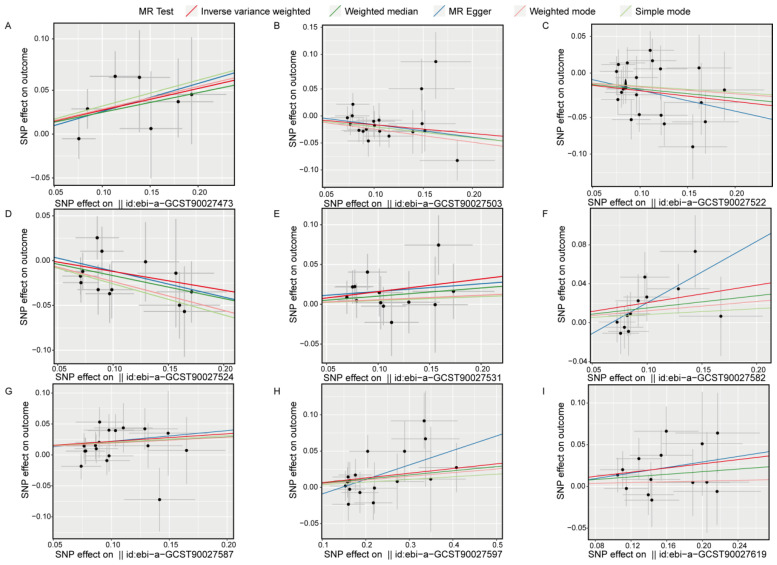
Scatter plots show the causal effects on the gut bacterial pathway abundance and the outcome (insomnia), respectively. (**A**), ebi-a-GCST90027473, GLUCONEO.PWY (gluconeogenesis I); (**B**), ebi-a-GCST90027503, P461.PWY (hexitol fermentation to lactate formate ethanol and acetate); (**C**), ebi-a-GCST90027522, PWY0.1479 (tRNA processing); (**D**), ebi-a-GCST90027524, PWY0.1586, (peptidoglycan maturation meso diaminopimelate containing); (**E**), ebi-a-GCST90027531, PWY.3001 (superpathway of L-isoleucine biosynthesis I); (**F**), ebi-a-GCST90027582, PWY.6519 (8-amino-7-oxononanoate biosynthesis I); (**G**), ebi-a-GCST90027587, PWY.6609 (adenine and adenosine salvage III); (**H**), ebi-a-GCST90027597, PWY.673 (starch degradation III); (**I**), ebi-a-GCST90027619, PWY.7254 (TCA cycle VII acetate producers). The dots represent SNP for causal estimation in the analysis direction; the different colored lines represent different MR analysis methods. IVW, inverse-variance weighted.

## Data Availability

All data used in this study are available in the public repository [OpenGWAS database https://gwas.mrcieu.ac.uk/, accessed on 10 July 2024]; [FinnGen database https://www.finngen.fi/en, accessed on 10 July 2024]. The code involved in the data analysis process can be obtained by contacting the corresponding author.

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
