# Peer review of "Metabolic Characteristics of Gut Microbiota and Insomnia: Evidence from a Mendelian Randomization Analysis"

_nutrients, 2024, doi:10.3390/nu16172943_

Round 1

Reviewer 1 Report

Comments and Suggestions for Authors

Metabolic Characteristics of Gut Microbiota and Insomnia: Evidence from a Mendelian Randomization Analysis

The authors can find my appraisal, suggestions, and concerns, section by section as follows:

 Introduction: The section is interesting, captivating, and well-written. The previous literature is well introduced, and the method is well presented. However, I suggest to be more specific about the hypotheses.

The methods are well explained. Step-by-step replicability is warranted. All the packages were explained, as well as the dataset, but a brief description is needed. However, what do you mean by causal effect? This needs to be stated in a better way.

 In the results, I advise removing the sentences between 207-212. These considerations are not “results” and may be better located in different sections.  Similarly, I advise to remove or move to another section the following statement “This provides a solid foundation for subsequent research, 277 prompting us to further explore how these gut bacterial pathway abundances specifically 278 influence the molecular mechanisms underlying insomnia.”.

The discussion is also interesting and I have no commentaries about. 

Author Response

Dear Professor,
We sincerely thank you for dedicating your valuable time and effort to our manuscript. Your detailed review and insightful comments are crucial for improving the quality of our paper. We deeply appreciate your expertise and constructive feedback. We have carefully read all the comments and suggestions you provided. In the accompanying detailed response, we will address each point you raised. We believe that by carefully considering and incorporating your suggestions, our manuscript will be significantly improved. Once again, thank you for your hard work and valuable contribution. Your professionalism and dedication are invaluable to the advancement of our academic community.
Yours sincerely,

Fuquan Xie

Comments 1: Introduction: The section is interesting, captivating, and well-written. The previous literature is well introduced, and the method is well presented. However, I suggest to be more specific about the hypotheses.

Response 1: 

Response 1: We sincerely appreciate your positive feedback on our introduction section. We are pleased that you found it interesting, captivating, and well-written. Your recognition of our efforts to effectively introduce the previous literature and present our method is greatly appreciated. We thank you for your constructive suggestion to be more specific about our hypotheses. We agree that clearly stated hypotheses are crucial for guiding the reader and framing the research. In light of your valuable feedback, we have revised our hypotheses to be more explicit and detailed.

  The updated hypotheses now read as follows: “This study hypothesizes that specific gut microbiome pathways may have a potential causal relationship with insomnia. Specifically, we expect that: First, single nucleotide polymorphisms (SNPs) closely associated with the abundance of gut microbiome pathways may influence the risk of insomnia. Second, certain microbial pathways (such as those related to neurotransmitter production or metabolism) may have stronger associations with insomnia risk than others. Third, these associations may be realized through altering the function of the gut-brain axis or influencing metabolic processes related to sleep regulation (such as gluconeogenesis). To test these hypotheses, this......”

  We believe these modifications provide a more concrete framework for our research, offering readers a clearer understanding of our study's objectives and expected outcomes. We are grateful for your insightful comment, which has helped us improve the clarity and specificity of our research framework. If you feel that further refinement is needed, we would be more than happy to make additional adjustments. Thank you once again for your thorough review and constructive feedback.

Comments 2: The methods are well explained. Step-by-step replicability is warranted. All the packages were explained, as well as the dataset, but a brief description is needed. However, what do you mean by causal effect? This needs to be stated in a better way.

Response 2: We greatly appreciate your thorough review of our methods section and your positive feedback regarding its explanatory nature and replicability. We are pleased that you found our description of the packages and dataset to be comprehensive.

We thank you for pointing out areas that require further clarification. We will address each of your comments as follows: 

  • Brief description of the dataset:
      We acknowledge that a more concise overview of our dataset would be beneficial. We propose to add the following brief description in the "2.1 Data Source" section (Line 112-117): "We used data from the OpenGWAS database (https://gwas.mrcieu.ac.uk/, accessed on 10 July 2024) on the abundance of 205 gut bacterial pathways (ebi-a-GCST90027446 to ebi-a-GCST90027650, sample size: 7,738) as exposure variables. Insomnia data from the R10 version of the FinnGen database (finngen_R10_F5_INSOMNIA, case: 4,801, control: 405,229) served as outcome variables."
  • Clarification of "causal effect":
        We sincerely appreciate your suggestion to better explain what we mean by "causal effect". You are correct that this term requires more precise definition in the context of our study. We propose to add the following explanation (Lines 135-142): "In this study, we use the term "causal effect" to describe the estimated impact of genetic variants associated with gut microbiome pathways on insomnia risk, as inferred through MR analysis. Specifically, a "causal effect" in this context suggests that variation in these genetic markers, which are proxies for gut microbiome pathway abundance, is associated with changes in insomnia risk. However, it's important to note that while MR provides evidence for potential causal relationships, it is subject to certain assumptions and limitations. In other words, the selected IVs must satisfy three core MR assumptions." We acknowledge that the term 'causal' should be interpreted cautiously, as our findings suggest potential causal pathways rather than definitively proving causation. We believe these additions will provide greater clarity on our dataset and our use of the term "causal effect". 

    We acknowledge that the term 'causal' should be interpreted cautiously, as our findings suggest potential causal pathways rather than definitively proving causation. We believe these additions will provide greater clarity on our dataset and our use of the term "causal effect". We are committed to ensuring the accuracy and transparency of our methods and results.

    Thank you again for your insightful comments. Your feedback is invaluable in helping us improve the clarity and rigor of our manuscript. If you feel that further elaboration or modification is needed, we would be more than happy to make additional adjustments.

Comments 3:  In the results, I advise removing the sentences between 207-212. These considerations are not “results” and may be better located in different sections.  Similarly, I advise to remove or move to another section the following statement “This provides a solid foundation for subsequent research, 277 prompting us to further explore how these gut bacterial pathway abundances specifically 278 influence the molecular mechanisms underlying insomnia.”.

Response 3: We sincerely appreciate your careful review of our results section and your thoughtful suggestions for improvement. Your attention to detail helps ensure that our manuscript maintains a clear and appropriate structure. We agree that maintaining a clear distinction between results and interpretative statements is crucial for scientific clarity.

Regarding your specific suggestions:

1. Removal of sentences between lines 207-212:
    We agree that these sentences do not strictly represent results and may be better suited elsewhere in the manuscript. Following your advice, we propose to remove these sentences from the results section. We will carefully review the content of these lines and, if appropriate, integrate relevant points into either the discussion section or the introduction, depending on their nature. This adjustment will help maintain the focus of our results section on empirical findings.

2. Relocation of the statement in lines 277-278:
    We appreciate your suggestion regarding the statement "This provides a solid foundation for subsequent research, prompting us to further explore how these gut bacterial pathway abundances specifically influence the molecular mechanisms underlying insomnia." We agree that this forward-looking statement is more suitable for the discussion section rather than the results. We propose to remove this sentence from results section. 

    These changes will help streamline our results section, ensuring it focuses solely on presenting our empirical findings. We are grateful for your insights, which will undoubtedly improve the overall structure and clarity of our manuscript. If you have any further suggestions or if you would like to review these changes once implemented, we would be more than happy to provide an updated draft for your consideration.

Thank you once again for your valuable feedback and your commitment to improving the quality of our research presentation.

Comments 4: The discussion is also interesting and I have no commentaries about. 

Response 4: We sincerely appreciate your positive feedback on our discussion section. We are pleased to hear that you found it interesting and that you have no further comments about it. 

Your assessment is encouraging and suggests that our discussion effectively synthesizes our findings, places them in the context of existing literature, and explores their implications. We strive to provide a comprehensive and insightful discussion that adds value to our research findings.

While you have no specific comments, we remain open to any future suggestions you might have for enhancing this section. Your expertise and perspective are invaluable to us, and we are committed to maintaining and improving the quality of our manuscript.

Thank you for your thorough review of our work, including this positive evaluation of our discussion section. Your feedback throughout the review process has been instrumental in refining our manuscript.

Reviewer 2 Report

Comments and Suggestions for Authors

The manuscript entitled "Metabolic Characteristics of Gut Microbiota and Insomnia: Evidence from a Mendelian Randomization Analysis" is an interesting article on a much debated topic. The relationship between insulin resistance and sleep apnea is known, which is exactly why the link between microbiota and insomnia is very interesting.  I am very interested in the topic, which links neurological disorders mediated by neurotransmitters, or sleep apnea, correlated with metabolic syndrome, or with any component thereof. All in all, the article is interesting, well written, corrected and addressed, but please rephrase the conclusions. Move the first sentence to discussion, and rephrase the rest more clearly.

For this reason, the study was very interesting and very clear for me.

1. The approach to insomnia is an original one, single nucleotide polymorphisms associated with gut bacterial pathway abundance were followed, which can determine an indirect but still very important link with insomnia, especially from the point of view of the gut-brain axis, as well as metabolically. This part was described both in the introduction (I was pleased to see that the gut-brain axis was addressed in the introduction) and the metabolic component, i.e. gluconeogenesis pathway in the discussions.

2. In 2023 yan li published in frontiers a work also on this topic, but without the gluconeogenic metabolic component, which for me is a very important part, and known for a longer time. I had nothing to comment on.

3. The methodology is developed correctly. The only part I would have merged is the statistical analysis part, in a single paragraph, but I considered it more explicit that way. Correct and very clear flowchart. I had no comments.

4. The conclusions can still be improved, as I wrote in the review. The first sentence rather belongs in discussions, and the conclusions are very general, the results should be emphasized more clearly. for example "Although removal of individual SNPs may affect the ranges of the effect estimates, the overall direction remains constant."

5. The figures are simple, (figure 2, figure 3) figure 4 is a representation of the range of variation of an effect size. I might not have made a figure, I would have just described, but the figure is correct, so I had no comment. Figure 5 is the most explicit, I personally like this the most, it is correct and represented according to statistical standards. I had nothing to comment on.

Author Response

Dear Professor,

We sincerely thank you for dedicating your valuable time and effort to our manuscript. Your detailed review and insightful comments are crucial for improving the quality of our paper. We deeply appreciate your expertise and constructive feedback.

We have carefully read all the comments and suggestions you provided. In the accompanying detailed response, we will address each point you raised. We believe that by carefully considering and incorporating your suggestions, our manuscript will be significantly improved.

Once again, thank you for your hard work and valuable contribution. Your professionalism and dedication are invaluable to the advancement of our academic community.

Yours sincerely,

Fuquan Xie

Comments 1: The approach to insomnia is an original one, single nucleotide polymorphisms associated with gut bacterial pathway abundance were followed, which can determine an indirect but still very important link with insomnia, especially from the point of view of the gut-brain axis, as well as metabolically. This part was described both in the introduction (I was pleased to see that the gut-brain axis was addressed in the introduction) and the metabolic component, i.e. gluconeogenesis pathway in the discussions.

Response 1: We sincerely appreciate your positive feedback on our approach to studying insomnia. We are pleased that you found our method of examining single nucleotide polymorphisms associated with gut bacterial pathway abundance to be original and valuable. Your recognition of the importance of the gut-brain axis in our study is greatly appreciated. We believe that this perspective offers a novel and potentially fruitful avenue for understanding the complex etiology of insomnia. By highlighting both the gut-brain axis in our introduction and the metabolic component (specifically the gluconeogenesis pathway) in our discussion, we aimed to provide a comprehensive framework for interpreting our findings. Your comment reinforces our belief that exploring the indirect yet significant links between gut microbiome-related genetic variations and insomnia can contribute meaningfully to the field. We are encouraged by your acknowledgment of the metabolic aspects of our study, particularly regarding the gluconeogenesis pathway. We thank you for your insightful comments and are glad that our approach resonates with experts in the field. Your feedback motivates us to further explore and refine this line of research in future studies.

Comments 2: In 2023 yan li published in frontiers a work also on this topic, but without the gluconeogenic metabolic component, which for me is a very important part, and known for a longer time. I had nothing to comment on.

Response 2: We greatly appreciate your bringing attention to the 2023 publication by Yan Li in Frontiers, which addresses a similar topic to our research. We thank you for highlighting this relevant work and for noting the key difference between our study and theirs. Indeed, our inclusion of the gluconeogenic metabolic component is a distinctive and crucial aspect of our research. We agree with your assessment that this is a very important part of the study. The gluconeogenesis pathway has been known for a longer time, as you correctly point out, and we believe its integration into our research provides a more comprehensive understanding of the complex relationships between gut microbiome, metabolism, and insomnia. By incorporating this metabolic component, our study builds upon existing knowledge and offers new insights into the multifaceted nature of insomnia. We aimed to bridge the gap between gut microbiome research and metabolic pathways, providing a more holistic view of the potential mechanisms underlying insomnia.

We are grateful for your recognition of this important aspect of our work. Your comment reinforces our belief in the value of integrating multiple physiological systems in our approach to understanding sleep disorders.

Thank you for your thorough review and for acknowledging the unique contributions of our research.

Comments 3: The methodology is developed correctly. The only part I would have merged is the statistical analysis part, in a single paragraph, but I considered it more explicit that way. Correct and very clear flowchart. I had no comments.

Response 3: We greatly appreciate your thorough review of our methodology and your positive feedback. We are pleased that you found our methodological approach to be correct and well-developed. Thank you for your comment regarding the statistical analysis section. We value your perspective on potentially merging this part into a single paragraph. Our decision to present it in its current format was aimed at enhancing clarity and facilitating easier comprehension of each analytical step. However, we appreciate your suggestion and understand the potential benefits of a more consolidated presentation. Your comment that you had no further remarks on this section is highly encouraging. It reinforces our confidence in the robustness and clarity of our methodological approach. We thank you for your constructive feedback and the time you've taken to carefully review our work. Your insights are invaluable in helping us refine and improve our research presentation.

Comments 4: The conclusions can still be improved, as I wrote in the review. The first sentence rather belongs in discussions, and the conclusions are very general, the results should be emphasized more clearly. for example "Although removal of individual SNPs may affect the ranges of the effect estimates, the overall direction remains constant." 

Response 4: We sincerely appreciate your insightful feedback on our conclusion section. We acknowledge that there is room for improvement, and we are grateful for your specific suggestions. We agree that the first sentence of our conclusion may be more suitable for the discussion section. We will revise our conclusion to focus more directly on the key findings and their implications.

The revised text of this section is as follows: "Although removing individual SNPs in the current MR analysis may affect the range of effect estimates, the overall trend of the MR analysis remains stable, providing robust evidence for our findings. The results indicate that key pathways such as carbohydrate metabolism, amino acid metabolism, and nucleotide metabolism demonstrated significant positive associations with the development of insomnia." We believe these changes will significantly strengthen our conclusion section, making it more focused and impactful. We are thankful for your careful review and constructive criticism, which will undoubtedly improve the quality of our manuscript. If you have any further suggestions or if you would like to see a draft of our revised conclusion, we would be more than happy to provide it for your review.

Response 5: We greatly appreciate your detailed review of our figures and your thoughtful feedback. Your comments are valuable in helping us understand how our visual representations are perceived by experts in the field. We respect your perspective and will consider whether a textual description might be more appropriate in future revisions or publications. We are particularly pleased that you found Figure 5 to be the most explicit and that you personally liked it the most. Your confirmation that it is correct and represented according to statistical standards is very reassuring. We put considerable effort into ensuring that this figure effectively communicates our findings while adhering to best practices in data visualization. Your overall assessment that you had no specific comments on the figures is encouraging. It suggests that, while there might be room for stylistic preferences, the figures effectively serve their purpose in communicating our research findings.

  We thank you for your careful examination of our visual presentations. Your feedback helps us ensure that our figures strike the right balance between simplicity, informativeness, and adherence to statistical standards.

Round 2

Reviewer 1 Report

Comments and Suggestions for Authors

The authors addressed all my concerns.